# Association between Osteoporosis and Cognitive Impairment during the Acute and Recovery Phases of Ischemic Stroke

**DOI:** 10.3390/medicina56060307

**Published:** 2020-06-23

**Authors:** Sang-Hwa Lee, So Young Park, Min Uk Jang, Yerim Kim, Jungyoup Lee, Chulho Kim, Yeo Jin Kim, Jong-Hee Sohn

**Affiliations:** 1Department of Neurology, Chuncheon Sacred Heart Hospital, Hallym University College of Medicine, 24253 Chuncheon, Korea; bleulsh@naver.com (S.-H.L.); gumdol52@naver.com (C.K.); yjhelena@hanmail.net (Y.J.K.); 2Department of Endocrinology and Metabolism, Kyung Hee University Hospital, 02247 Seoul, Korea; malcoy@hanmail.net; 3Department of Neurology, Dongtan Sacred Heart Hospital, Hallym University College of Medicine, 18450 Hwaseong, Korea; mujang@gmail.com; 4Department of Neurology, Kangdong Sacred Heart Hospital, Hallym University College of Medicine, 05355 Seoul, Korea; brainyrk@gmail.com; 5Departement of Emergency Medicine, Graduate School, Kyung Hee University, 02453 Seoul, Korea; doctormunubal@hanmail.net

**Keywords:** osteoporosis, bone mineral density, cognitive impairment, white matter disease, ischemic stroke

## Abstract

*Background and objectives:* Little is known about the effect of osteoporosis on cognitive function in the acute and recovery phases of stroke. Early bone mineral density assessments during acute stroke may be a useful marker of cognitive function. We evaluated the effect of osteoporosis on cognitive function at the early and recovery phase of ischemic stroke in patients aged >50 years. *Materials and Methods*: We retrospectively examined consecutive patients with acute stroke hospitalized between 2016 and 2018. Osteoporosis was defined as a T-score <–2.5 for the femoral neck or lumbar spine bone mineral density. The primary outcome was cognitive impairment measured by the Korean Mini-Mental State Examination in the acute phase and recovery phase of ischemic stroke. *Results:* Of the 260 included subjects (107 men and 153 women), 70 (26.9%) had osteoporosis. Cognitive impairment was more severe in the osteoporosis group than in the non-osteoporosis group (30.5% versus 47.1%, *p* = 0.001). After the recovery phase of stroke, the proportion of patients with cognitive impairment remained higher in the osteoporosis group. The multivariate analysis revealed a correlation between a low femoral neck bone mineral density and severe cognitive impairment in the acute and recovery phases of stroke (adjusted odds ratio (OR) 4.09, 95% confidence interval (CI) 1.11–15.14 in the acute phase, and adjusted OR 11.17, 95% CI 1.12–110.98 in the recovery phase). *Conclusions:* Low bone mineral density is associated with poor cognitive function in the acute and recovery phases of stroke.

## 1. Introduction

Several studies have investigated the relationship between low bone mineral density (BMD) and ischemic stroke, and have shown that low BMD represents a potential risk factor for stroke and affects the long-term outcome of ischemic stroke [1,2,3,4]. Nonetheless, researchers have not clearly determined whether osteoporosis is a marker for stroke. However, a potential complex causal link between stroke and osteoporosis has been reported. Since ischemic stroke can result in physical disability and cognitive impairment [5], and 40% of patients with stroke present with preexisting osteoporosis [6], a study designed to evaluate the effect of osteoporosis on cognitive impairment after ischemic stroke would be interesting. Cognitive function is known to be influenced by the time of stroke occurrence. Hence, the treatment strategy of stroke may vary depending on the early and delayed periods of stroke.

Recently, BMD was reported to be associated with microvascular diseases, such as cerebral white matter disease (WMD): a lower BMD indicates a more severe WMD [7,8]. Additionally, WMD is a well-known risk factor for dementia in several studies [9]. Hence, we postulated that a low BMD may be associated with cognitive impairment.

In this study, we investigated the relationship between BMD and cognitive function in patients with ischemic stroke during the acute and recovery phases, using a single-center registry database in an attempt to reveal a new prognostic factor for cognitive impairment in patients with stroke.

## 2. Materials and Methods

### 2.1. Study Population

We included consecutive patients with acute ischemic stroke who were hospitalized at the Chuncheon Sacred Heart Hospital between January 2016 and June 2018. All patients over 50 years of age were subjected to routine bone densitometry within 7 days of stroke onset. We excluded patients who had a surgical or medical history affecting bone turnover (traumatic fracture; spinal surgery; liver, renal, or thyroid disease; and hormonal agent use) and a pre-stroke modified Rankin Scale (mRS) score of >3. All females enrolled in this study were menopausal. Unconscious patients whose cognitive function was unable to be assessed were also excluded. We established three databases to investigate the effects of osteoporosis on cognitive function and WMD. Because patients with severe neurological deficits were unable to perform or refuse cognitive function tests, we established three separate databases: database 1 excluded subjects who were not able to complete the Korean Mini-Mental State Examination (K-MMSE), evaluating the effect of osteoporosis on cognitive function during hospitalization; database 2 excluded subjects who were not able to complete the K-MMSE at 1 year after stroke onset; and database 3 included all eligible subjects with WMD (Figure 1). Bone loss is accelerated during the first year after stroke [10], and the risk for post-stroke cognitive impairment is known to be highest within the first year [11]. Hence, we assessed the BMD at the early phase of stroke and the cognitive function test at hospitalization and 1 year after a stroke.

### 2.2. Data Collection and Outcome Measures

The following data were obtained directly from the registry database: (1) demographic characteristics (age and sex); (2) stroke risk factors and medical history (previous stroke, hypertension, diabetes mellitus, hyperlipidemia, atrial fibrillation, current smoking status, and prior antithrombotic drug use); (3) clinical characteristics and acute stroke treatment (initial National Institute of Health Stroke Severity Scale (NIHSS) score, which is a tool to quantify the degree of impairment after stroke (the maximum possible score is 42, with the minimum score being a 0); the ischemic stroke mechanism, according to the TOAST criteria with some modifications (small vessel occlusion, large artery atherosclerosis, cardioembolism and others) [12]; educational level (years); body mass index; and systolic blood pressure); (4) laboratory data (white blood cell count and hemoglobin, creatinine, initial random glucose, low-density lipoprotein and glycated hemoglobin levels); and (5) functional status, defined by the mRS score (0 = no symptoms; 1 = no significant disability; 2 = slight disability, able to look after own affairs without assistance; 3 = moderate disability, requires some help, but able to walk unassisted; 4 = moderately severe disability, unable to attend to own bodily needs without assistance; 5 = severe disability, requires constant nursing care and attention, bedridden; 6 = dead) at discharge and at 3 months [13].

The primary outcome measure was the severity of cognitive impairment, and the secondary outcome measure was the extent of WMD stratified by BMD T-scores of <−2.5 and ≥−2.5.

### 2.3. BMD Measurements

The BMD at the lumbar spine and femoral neck was estimated using dual-energy X-ray absorptiometry (Horizon W DXA system, HOLOGIC, United States). The coefficient of variation was approximately 1% for the BMD of the lumbar spine and femoral neck. The lumbar spine BMD was measured as the mean BMD from the first to the fourth lumbar vertebra. The femoral neck BMD was measured on only the right side. Osteoporosis was defined as a BMD at the lumbar spine or femoral neck that was at least 2.5 standard deviations (SDs) below the reference mean (T-score <−2.5) [14]. The volumetric BMD (g/cm^2^) was also measured at both the lumbar spine and femoral neck. The BMD was measured in all subjects during hospitalization.

### 2.4. Cognitive Function Test

The K-MMSE was used to assess the cognitive function of the subjects [15]. The K-MMSE is the most widely applied cognitive function screening tool used in clinical research in Korea. The initial K-MMSE was administered by an expert neuropsychologist within 7 days of stroke onset. At 1 year after ischemic stroke onset, which was considered the recovery phase of stroke, a follow-up K-MMSE was administered to each patient. We categorized cognitive impairment as follows: no cognitive impairment (K-MMSE score of 24–30 points), mild cognitive impairment (K-MMSE score of 18–23 points), and severe cognitive impairment (K-MMSE score of 0–17 points) [16].

### 2.5. Assessment of WMD

All subjects were subjected to brain magnetic resonance imaging (MRI). The Fazekas scale was used to quantify the periventricular and deep subcortical white matter on the MRI fluid attenuation inversion recovery images, and assess the presence of a WMD [17]. The Fazekas scale is recommended to assess the extent of WMD, due to its simplicity and applicability in MRI. The severity of WMD was defined as follows: no punctate white matter foci (Fazekas score = 0), multiple punctate white matter foci (Fazekas score = 1, mild), incipient confluence or bridging of punctate foci (Fazekas score = 2, moderate), and confluent white matter areas (Fazekas score = 3, severe). Two expert neurologists (S.-H. Lee and M.U. Jang) assessed the WMD severity. The interobserver reliability of the WMD scale was acceptable (kappa coefficient = 0.81).

### 2.6. Statistical Analysis

The summary statistics are presented as the numbers of subjects (percentage) for categorical variables and as the means ± SDs or medians for continuous variables. Group comparisons were performed using Pearson’s chi-square test or Fisher’s exact test for categorical variables, and Student’s *t*-test or the Mann–Whitney U test for continuous variables, as appropriate.

In this study, we separately analyzed the effect of BMD of the femoral neck (Model 1) and BMD of the lumbar spine (Model 2) on the outcomes. The osteoporosis group was a combination of the BMD T-score <−2.5 of the femur neck and lumbar spine. With respect to the primary and secondary outcome measures, the osteoporosis and non-osteoporosis groups were compared using Pearson’s chi-squared test, and independent effects of osteoporosis on those outcome measures were analyzed using multivariate regression analysis. The multivariate analysis was adjusted for variables selected based on *p*-values <0.1 in the comparisons of groups stratified according to osteoporosis severity, and clinically plausible associations with each outcome variable were determined. Moreover, an interaction analysis was performed as to whether WMD modified the association between BMD and cognitive impairment.

We performed a multivariate analysis to evaluate the impact of BMD on cognitive function during the acute and recovery phases after stroke, using databases 1 and 2, as well as the impact of BMD on WMD using database 3.

As a sensitivity analysis, we evaluated the impact of osteoporosis on cognitive function, as defined by the Montreal Cognitive Assessment (MoCA), using the van Steenovan MoCA–MMSE conversion scale. We divided the MoCA scores into three groups (MoCA score of 0~10, 11~17, and 18~30 points) [18,19,20]. The sensitivity of the association between the volumetric BMD (g/cm^2^) and the K-MMSE score was investigated using a linear regression analysis. All data analyses were performed with IBM SPSS version 21.0 software (IBM Corporation, Armonk, NY, United States). Crude and adjusted odds ratios (ORs) and 95% confidence intervals (CIs) were estimated.

Because of the anonymity of the study subjects and the minimal risk to the patients, the collection of data without informed consent was approved by the local institutional review board (IRB) of Chuncheon Sacred Heart Hospital before the study commenced (IRB No. 2018-08-020).

## 3. Results

### 3.1. Baseline Characteristics

Over the study period, 302 patients (125 males and 177 females, aged 73.3 ± 10.5 years) with acute ischemic stroke underwent bone densitometry (database 3, Appendix A). Of the 302 patients, 260 patients were assessed using the K-MMSE (database 1; 107 males and 153 females, aged 72.9 ± 10.5 years). Of the 260 patients, 239 patients were assessed with the follow-up K-MMSE in the recovery phase of stroke (database 2; 98 males and 141 females, aged 73.2 ± 10.5). Seventy patients (26.9%) in database 1 were diagnosed with osteoporosis (BMD T-score <−2.5), 38 (14.6%) had a femoral neck BMD T-score of <−2.5, 51 (19.6%) had a lumbar spine BMD T-score of <–2.5, and 19 had both a femoral neck and lumbar spine BMD T-score of <−2.5. The baseline clinical characteristics stratified by the BMD at the femoral neck and lumbar spine are shown in Table 1, as well as Appendix A. Patients in the osteoporosis group tended to be older and were more likely to be female, have higher creatinine levels, and not be current smokers. However, the stroke severity measured using the NIHSS and the stroke subtypes were not different between the osteoporosis and non-osteoporosis groups. In addition, the functional status (mRS) at hospitalization and at 3 months did not differ between the osteoporosis and non-osteoporosis groups.

### 3.2. Osteoporosis and Cognitive Function

The osteoporosis group had more cognitive impairment (K-MMSE score < 24) than the non-osteoporosis group (81.4% versus 57.4%; *p* < 0.001). During the acute phase of ischemic stroke, the proportion of severe cognitive impairment (K-MMSE score of 0 to 17) in the osteoporosis group was higher that the non-osteoporosis group (47.1% versus 30.5%; *p* = 0.001). Additionally, the cognitive impairment of the group with femoral neck osteoporosis was significantly more severe than the group without femoral neck osteoporosis (45.1% versus 32.5%; *p* = 0.002), whereas the cognitive impairment of the group with lumbar spine osteoporosis was not different from the group without lumbar spine osteoporosis (Figure 2A). During the recovery phase of ischemic stroke, the proportion of patients with a severe cognitive impairment in the osteoporosis group remained higher than the non-osteoporosis group (44.3% versus 29.8%; *p* = 0.002). The proportion of patients with a severe cognitive impairment in the group with femoral neck osteoporosis during the recovery phase of stroke remained higher than the group without femoral neck osteoporosis (51.5% versus 30.6%; *p* = 0.01; Figure 2B). The extent of WMD was more likely to be moderate to severe in the osteoporosis group than in the non-osteoporosis group (73.2% versus 57.9%; *p* = 0.02). Furthermore, the group with femoral neck osteoporosis had significantly more severe WMD than the group without femoral neck osteoporosis (85.5% versus 57.9%; *p* = 0.001), but a similar difference was not observed for patients with lumbar spine osteoporosis (Figure 3). With respect to the results according to the MoCA scores, the proportion of patients with lower MoCA scores (0~10 points) was relatively higher in the group with femoral neck osteoporosis in both the acute and recovery phases of stroke (31.1% versus 57.9%, *p* = 0.001 in the acute phase, and 47.1% versus 78.8%, *p* = 0.002 in the recovery phase; Appendix A).

### 3.3. Association between Osteoporosis and Cognitive Function

The multivariate analysis revealed a correlation between osteoporosis and mild cognitive impairment in the acute phase, but not with severity of cognitive impairment during the recovery phase (defined as 1 year after stroke) of ischemic stroke (Table 2, Appendix A). Osteoporosis of the femoral neck was associated with a severe cognitive impairment (adjusted OR = 4.09, 95% CI = 1.11–15.14, *p* = 0.035 in the acute phase; adjusted OR = 11.17, 95% CI = 1.12–110.98, *p* = 0.04 in the recovery phase), whereas osteoporosis of the lumbar spine was not associated with cognitive impairment. The multivariate analysis did not reveal a correlation between osteoporosis assessed by measuring the total BMD and the extent of WMD. However, osteoporosis of the femoral neck increased the risk of moderate and severe WMD. Osteoporosis of the lumbar spine did not increase the extent of WMD (Table 3 and Appendix A). The mediator effect observed in the interaction analysis showed that the association between BMD and cognitive impairment was significantly modified by WMD during the recovery phase of ischemic stroke (*p*-values for interaction < 0.001, Appendix A). In the sensitivity analysis, osteoporosis of the femoral neck was associated with a lower MoCA score in both the acute and recovery phases of stroke (adjusted OR (95% CI) = 3.44 (1.03–11.51) in the acute phase, and 12.49 (1.30–119.96) in the recovery phase; Appendix A).

The linear regression sensitivity analysis revealed a positive correlation between the volumetric BMD at the femoral neck and increasing K-MMSE scores during both the acute and recovery phases of ischemic stroke (β = 0.23, *R*^2^ = 0.26, *p* < 0.001 in the acute phase; β = 0.19, *R*^2^ = 0.36, *p* = 0.01 in the recovery phase). However, the volumetric BMD at the lumbar spine was not significantly correlated with the K-MMSE score in either phase of ischemic stroke (Appendix A).

## 4. Discussion

We are the first group to show that osteoporosis of the femoral neck is an independent predictor of cognitive impairment in the acute and recovery phases of ischemic stroke. Additionally, osteoporosis of the femoral neck increases the risk of either moderate or severe WMD. These results have significant clinical implications. Despite recommendations to measure BMD in patients with acute stroke presenting risk factors for osteoporosis, BMD is not evaluated routinely in most stroke centers [21,22]. Approximately 40% of patients with hip fractures have a cognitive impairment [23]. Because stroke could lead to an increased risk of cognitive impairment, the early detection and treatment of osteoporosis might promote improved cognitive function in patients with stroke.

In the present study, the percentages of patients presenting a cognitive impairment based on a K-MMSE score <24 were 63.8% (28.8% with a mild cognitive impairment and 35.0% with a severe cognitive impairment) during the acute phase and 63.6% (30.1% with a mild cognitive impairment and 33.5% with a severe cognitive impairment) during the recovery phase of ischemic stroke. Although pre-stroke cognitive status was not available in the present study, the proportion of patients presenting with a cognitive impairment during the recovery phase of stroke is comparable to previous studies reporting a prevalence of cognitive impairment after stroke ranging from 24% to 70% [24]. Regarding the prevalence of post-stroke cognitive impairment, findings from this study support emerging evidence of the importance of improving strategies designed for the secondary prevention of post-stroke cognitive impairment. In addition, the associations between osteoporosis and MoCA scores in the sensitivity analysis support our main findings. Based on our results, we cautiously recommend that clinicians use BMD as a new marker of cognitive impairment after stroke.

Our study reveals the association between osteoporosis and cognitive impairment, but did not demonstrate causality. Osteoporosis is known to affect microvascular diseases, such as cerebral WMD. The same mechanisms of dysregulated proteolytic processing and ion and electron transport have been described in genetic and laboratory studies of both cerebral WMD and bone osteoporosis [25,26]. In addition, WMD is a strong predictor of vascular cognitive impairment and dementia [27,28,29,30]. Notably, the mediator effect analysis shows that WMD is potentially attributable to BMD and cognitive impairment during the recovery phase of acute ischemic stroke. Based on these laboratory and statistical results, we carefully postulated that osteoporosis affects cognitive function by inducing microvascular damage during WMD progression. However, we should be cautious when generalizing this finding, since we did not consider unmeasured confounding factors.

Interestingly, femoral neck BMD, but not lumbar spine BMD, was associated with the severity of cognitive impairment and WMD. In previous studies of cardiovascular disease, femoral neck BMD was significantly associated with severe coronary atherosclerosis and an increased risk of coronary artery disease [31,32]. This association was explained by the greater likelihood that spinal BMD will be affected by drugs, medical conditions, and degenerative arthritis in elderly people than femoral neck BMD [33]. The correlation between the volumetric femoral neck and lumbar spine BMDs and the MMSE score, as well as the association between the femoral neck BMD and MoCA score in this study strongly support this hypothesis. Therefore, femoral neck BMD potentially represents a more useful marker for predicting cognition and WMD than lumbar spine BMD. However, since both femoral neck and lumbar spine BMD were associated with a poor outcome in the previous analysis of cerebrovascular disorders, and other studies measured only the femur neck BMD to investigate stroke risk [1,3], we should use caution when generalizing these results. Also, pre-stroke BMD was not available in our study, and the lumbar spine has known to be less sensitive to mechanical stimulation. Hence, we should be careful about clinically interpreting this result. Nonetheless, these results could prompt researchers to identify a reasonable BMD site for predicting cognition in patients with stroke. Further studies are needed to clarify this issue.

Osteoporosis is associated with atherosclerosis [34,35,36,37], but this association remains controversial [38]. Most of these studies have focused on coronary artery disease. In the study, large artery atherosclerosis tended to be an uncommon ischemic stroke mechanism, whereas the incidence of stroke caused by small vessel occlusion tended to be high in patients with osteoporosis. Although osteoporosis is known to share risk factors with atherosclerosis, studies examining the relationship between atherosclerosis and osteoporosis in patients with ischemic stroke are rare. Regarding the ischemic stroke mechanism, the proportion of patients presenting with atherosclerosis who are diagnosed with ischemic stroke is relatively low (ranging from 9.3% to 20.9% in patients with ischemic stroke) compared to the proportion of patients presenting with atherosclerosis who are diagnosed with coronary artery disease. Thus, osteoporosis may exhibit a stronger correlation with microvascular disease than with atherosclerosis in subjects with a brain infarction. However, this conclusion should be interpreted with caution, due to the small sample size of our study.

Although this study is the first to show an association between a low BMD and cognition in patients with stroke, it has several limitations. First, this study was a single-center study with a relatively small number of subjects, despite the use of a registry database consecutively collecting patients’ data. Further multicenter studies assessing cognitive function should be established to confirm our results. Second, since the cognitive function test was not administered prior to stroke, the patients’ pre-stroke cognitive statuses were unavailable. Further study focused on this issue should use informant assessment, such as the Informant Questionnaire for Cognitive Decline (IQCODE), which could be a useful tool for assessing pre-stroke cognitive status [39]. However, the aim of our study was to determine the trend in cognitive decline during the acute and recovery phases in both the osteoporosis and non-osteoporosis groups, and to compare the impact of osteoporosis on cognitive function in those two phases of stroke. Also, we proposed that WMD had an important role in the effect of BMD on cognitive decline. Since we compared the effect of BMD on cognitive impairment in both the acute and recovery phases of stroke, our results could have important clinical implications. Third, in the present study, the mean age of all the subjects was high (72.9 ± 10.5 years), and the osteoporosis group tended to be older than the non-osteoporosis group, which may have affected the outcome. Post-stroke depression also could affect the cognition of the subjects. After reviewing the Geriatric Depression Scale (GDS), which was administered at the same time as the MMSE, the GDS scores were not different between the two groups (data were not shown). Since we adjusted for age and sex in all multivariate models, and found no difference in GDS scores between the osteoporosis group and the non-osteoporosis group, we assume that the impacts of age and depression on the outcomes have been attenuated in this study. Fourth, although we controlled for known measurable confounders in the multivariate analysis, unmeasured confounding factors (physical activity and gait speed prior to stroke, acute stressors like emotional stress and depression due to hospitalization or stroke symptoms, etc.) could remain, and may threaten the generalizability of our results. Although scores for physical activity and gait speed prior to stroke were not available in this study, we excluded subjects with independent functional statuses (mRS > 3), and the 3 month functional status scores (mRS > 2) were not different between the two groups. In addition, acute stressful situations can affect the cognitive status of a patient in the acute phase of stroke. However, we aimed to evaluate the impact of osteoporosis on cognition 1 year after stroke onset, which should therefore be free from confounding by the effects of acutely stressful situations. We additionally found that the adjusted ORs for the effect of osteoporosis on cognition tended to increase in the recovery phase (Table 2). Last, the effects of the stroke itself (lesion locations and infarct volume, etc.) on delayed cognitive impairment were not considered in the present study. However, the locations of stroke lesions were not significantly different between the osteoporosis group and the non-osteoporosis group. Although the infarct volume was not available in our study, we cautiously suggest that a lack of difference in the proportions of lesion locations could increase the validity of our results.

## 5. Conclusions

A low BMD is associated with poor cognitive function in ischemic stroke patients. Although we did not determine causality between cognition and BMD, this study might allow clinicians to consider the association between osteoporosis and cognitive decline in patients with ischemic stroke. Also, we cautiously suggest that the extent of WMD could be a potential mediator of the relationship between osteoporosis and cognitive function in stroke patients. The underlying mechanisms and prognostic predictors of osteoporosis and cognition require confirmation in further studies.

## Figures and Tables

**Figure 1 medicina-56-00307-f001:**
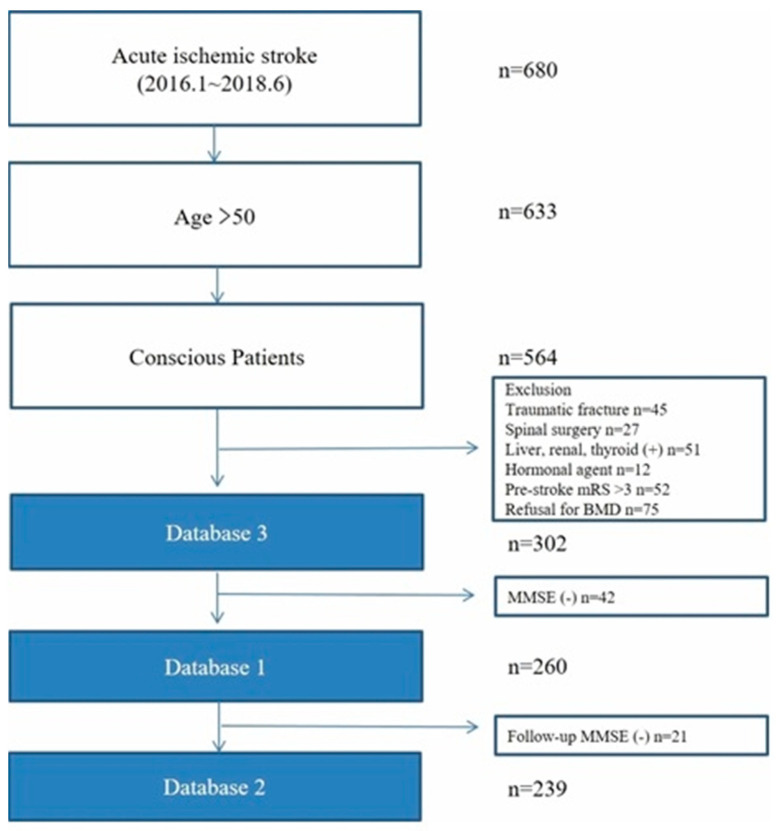
Flow chart of the study. Database 1: excluding subjects without MMSE at hospitalization; database 2: excluding subjects without MMSE at 1 year after stroke. MMSE: Mini-Mental State Examination.

**Figure 2 medicina-56-00307-f002:**
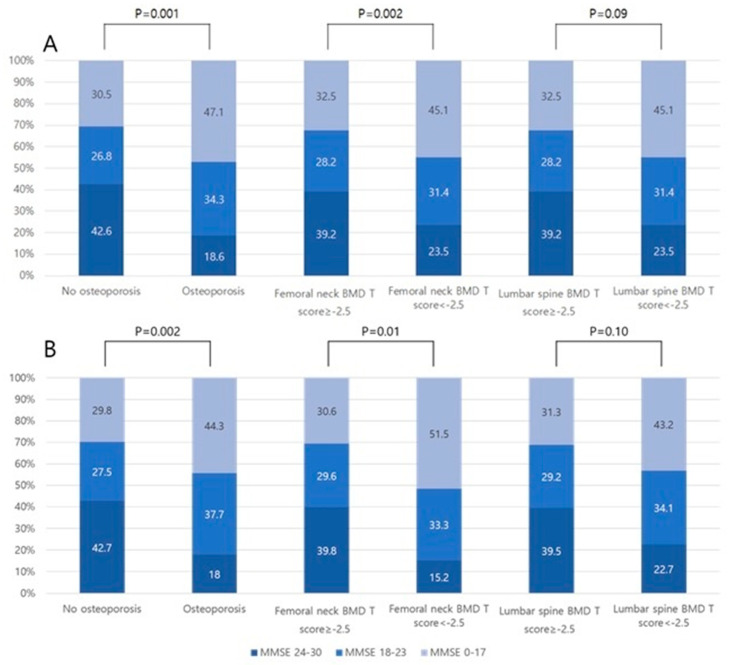
Cognitive impairment in groups stratified according to BMD (total BMD, femoral neck BMD, and lumbar spine BMD) during the acute phase (**A**) and recovery phase (**B**) of ischemic stroke. Abbreviations: BMD, bone mineral density; MMSE, Mini-Mental State Examination.

**Figure 3 medicina-56-00307-f003:**
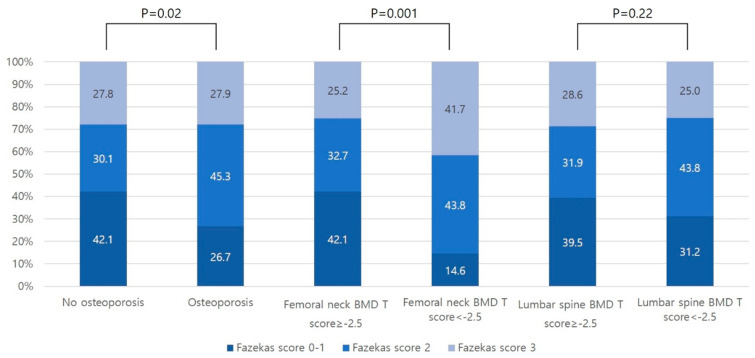
Extent of WMD in groups stratified according to BMD (total BMD, femoral neck and lumbar spine BMD). WMD: white matter disease.

**Table 1 medicina-56-00307-t001:** Baseline characteristics of the osteoporosis group and the non-osteoporosis group (using database 1, *n* = 260).

	Non-Osteoporosis (*n* = 190)	Osteoporosis (*n* = 70)	*p*-Value
Age, years (SD)	71.5 (10.9)	76.7 (8.4)	0.06 ^b^
Female (%)	90 (47.4)	63 (90.0)	<0.001 ^a^
BMI, kg/m^2^ (SD)	24.4 (4.0)	22.9 (3.6)	0.90 ^b^
Initial NIHSS score (%)			0.70 ^a^
0–4	132 (69.5)	48 (68.6)	
5–11	45 (23.7)	19 (27.1)	
>11	13 (6.8)	3 (4.3)	
Stroke mechanisms (%)			0.44 ^a^
SVO	50 (26.3)	23 (32.9)	
LAA	72 (37.9)	21 (30.0)	
CE	34 (17.9)	10 (14.3)	
Other	34 (17.9)	16 (14.3)	
Prior stroke (%)	46 (24.2)	20 (28.6)	0.52 ^a^
Hypertension (%)	121 (63.7)	49 (70.0)	0.38 ^a^
Diabetes mellitus (%)	73 (38.4)	19 (27.1)	0.11 ^a^
Hyperlipidemia (%)	28 (14.7)	9 (12.9)	0.84 ^a^
Current smoker (%)	33 (17.4)	1 (1.4)	0.001 ^a^
Atrial fibrillation (%)	38 (20.0)	12 (17.1)	0.72 ^a^
Prior antithrombotic agent use (%)	82 (43.2)	29 (41.4)	0.89 ^a^
Education, years (SD)	6.9 (4.3)	6.3 (4.1)	0.06 ^b^
Lesions			0.64 ^a^
Supratentorial	135 (71.1)	52 (74.3)	
Infratentorial	55 (28.9)	18 (25.7)	
mRS >2 at discharge (%)	47 (24.7)	19 (27.1)	0.75 ^a^
mRS >2 at 3 months (%)	64 (33.7)	25 (35.7)	0.77 ^a^
MMSE score (IQR)	22 (14-27)	18 (11-22)	<0.001 ^c^
LDL, mg/dL (SD)	98.2 (34.5)	89.4 (28.0)	0.09 ^b^
Creatinine, mg/dL (SD)	0.97 (0.37)	1.11 (0.97)	0.03 ^b^
HbA1c, % (SD)	6.3 (1.4)	6.2 (1.3)	0.58 ^b^

Abbreviations: SD, standard deviation; BMI, body mass index; NIHSS, National Institute Health of Stroke Scale; SVO, small vessel occlusion; LAA, large artery atherosclerosis; CE, cardioembolism; mRS, modified Rankin Scale; MMSE, mini-mental state examination; WBC, white blood cell; LDL, low-density lipoprotein; HbA1c, glycated hemoglobin; SBP, systolic blood pressure; ^a^ calculated using the chi-square test; ^b^ calculated using Student’s *t*-test; ^c^ calculated using Mann–Whitney U test.

**Table 2 medicina-56-00307-t002:** Multivariate analysis: association between BMD and cognitive impairment (analyzed using databases 1 and 2).

	Mild Cognitive Impairment vs. No Cognitive Impairment		Severe Cognitive Impairment vs. No Cognitive Impairment	
	Adjusted OR (95% CI) in the Acute Phase	Adjusted OR (95% CI) in the Recovery Phase	Adjusted OR (95% CI) in the Acute Phase	Adjusted OR (95% CI) in the Recovery Phase
Total BMD T-score < −2.5 ^a^	2.64 (1.05–6.61) *	2.90 (0.80–10.52)	2.22 (0.84–5.86)	2.55 (0.69–9.51)
Femoral neck BMD T-score < −2.5 ^b^	3.09 (0.87–10.99)	4.28 (0.43–42.21)	4.09 (1.11–15.14) *	11.17 (1.12–110.98) *
Lumbar spine BMD T-score < −2.5 ^a^	1.07 (0.40–2.84)	2.94 (0.68–12.61)	0.92 (0.34–2.50)	1.53 (0.35–6.70)

Abbreviations: BMD, bone mineral density; OR, odds ratio; CI, confidence interval. ^a^ Adjusted for age, female sex, years of education, diabetes mellitus status, current smoking status, low-density lipoprotein level, creatinine level, and the Fazekas score; ^b^ adjusted for age, female sex, years of education, prior stroke status, current smoking status, low-density lipoprotein level, creatinine level, and the Fazekas score; * means statistically significant.

**Table 3 medicina-56-00307-t003:** Multivariate analysis: association between BMD and WMD (analyzed using database 3).

	Moderate WMD vs. Normal-to-Mild WMD		Severe WMD vs. Normal-to-Mild WMD	
	Adjusted OR (95% CI)	*p*-Value	Adjusted OR (95% CI)	*p*-Value
Total BMD T-score < −2.5 ^a^	1.96 (0.98–3.90)	0.06	1.06 (0.509–2.26)	0.88
Femoral neck BMD T-score<−2.5 ^b^	2.71 (1.03–7.15)	0.04	3.13 (1.14–8.60)	0.03
Lumbar spine BMD T-score < −2.5 ^a^	1.33 (0.64–2.76)	0.45	0.74 (0.33–1.67)	0.47

Abbreviations: BMD, bone mineral density; WMD, white matter disease; OR, odds ratio; CI, confidence interval. ^a^ Adjusted for age, female sex, years of education, diabetes mellitus status, current smoking status, and low-density lipoprotein and creatinine levels; ^b^ adjusted for age, female sex, years of education, prior stroke status, current smoking status, and low-density lipoprotein and creatinine levels.

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
