# Peer review of "Association between Osteoporosis and Cognitive Impairment during the Acute and Recovery Phases of Ischemic Stroke"

_medicina, 2020, doi:10.3390/medicina56060307_

Round 1

Reviewer 1 Report

In the current paper, Lee et al. investigate the association between bone mineral density and cognition in a cohort of patients recovering from ischemic strokes. The study is novel and has several interesting findings. The paper itself is relatively well written. There are however some minor concerns that the reviewer would suggest that the authors consider to improve clarity, as well as, some potential analysis suggestions that may help address some of the conclusions: 

Introduction

  • Can the authors be more specific about what specific "long-term outcome of ischemic stroke" that BMD affects? It is return to physical function?

Methods

  • did the authors include bisphosphonate use in their screening criteria? Is it known whether any of the patients included in the model are taking any drugs for osteoporosis?
  • Can you clarify the units or the levels for each of the variables used in your models? For example, are previous stroke and hypertension all modeled as two level, yes/no variables? Also, how were all of these variables assessed? Questioning the patients or chart review?
  • Can the authors clarify in the methods and results clarify the stratification strategy? You state that the multivariate modeling is stratified by osteoporosis severity, I assume that this means that osteoporosis was separated into a two level variable of tscore<-2.5 and tscore > 2.5? 
  • Can you also include units for variables like BMI?
  • The methods don't include mention of all of the variables in Table 1. For example can the author include a description of NIHSS score and stroke subtype in the methods?
  • Just to clarify, the mediation analysis test the interaction of BMD and WMD (i.e BMDxWMD)?
  • What is the unit of change of BMD used in the linear regression models? It is the t-score or g/cm2?
  • When was WMD measured? During the acute phase, the recovery phase, or both? 

Results

  • 3.2 "The osteoporosis group likely had more..." the results show that this is significant. Would recommend removing the term likely.
  • Check the image quality of figure 2
  • In tables 2 and 3, can the authors please label which phase of the stroke they are presenting data from? For example, I believe table 2 is in the acute phase, but the table doesn't clearly present this. 
  • In table 2 it would be nice to show when the relationship is significant. It is mentioned in the text, but it if the authors could include the p-value, or give some other indication, such as bolding or adding a star or asterix, it would help the readers follow along.
  • What is the relationship between cognition in the acute phase and recovery phase? For example, if someone is cognitively impaired in the acute phase, do they have a very high likelihood of being cognitively impaired in the recovery phase as well? If so, did the authors adjust for acute cognition in any of the models run in the recovery phase? Would it be possible to include change in cognition between acute and recovery to test whether osteoporosis or BMD affected the trajectory? 

Discussion

  • The authors refer WMD as a potential mediator in the discussion, but it doesn't appear that Fazekas scale is associated with cognition in models including BMD (Tables s5 and s6). If it were mediating the association between BMD and cognition, wouldn't the authors first run models showing that BMD is significantly associated with cognition, then run models showing that when Fazekas score is included, the association between BMD and cognition is no longer significant or considerably attenuated? Unless I am missing something, the results presented in Tables 2 and 3 seem to suggest that BMD contributes to both WMD and cognition and the contribution of BMD to cognition is independent of WMD.  
  • Line 46 - do the authors have any specific unmeasured confounding factors that they believe would would help with the generalizability of their findings? 
  • In the limitations, the authors note that there population was older, which is true, but they don't comment on the fact that is includes more female patients. 
  • The authors reference an association between osteoporosis and cognitive decline, but unless I missed it there was not time-dependent variable to get at cognitive decline, wasn't the analysis primarily cross-sectional at two time points separately? I would recommend change to an association between osteoporosis and cognition.
  • The conclusions seem too broad. For example A low BMD is associated with poor cognitive function should probably include "in ischemic stroke patients."

Author Response

Reviewer reports:

Reviewer#1

Introduction

  1. Can the authors be more specific about what specific "long-term outcome of ischemic stroke" that BMD affects? It is return to physical function?

Response: Several studies indicated that osteoporosis could increase the stroke risk and associated with poor functional status and death after stroke onset. The potential mechanisms were proposed but still have remained unclear. Unlike previous studies, we evaluated how low BMD affects cognitive function after stroke onset and proposed the potential causal link between BMD and cognitive function via interaction effect of WMD.

Methods

  1. did the authors include bisphosphonate use in their screening criteria? Is it known whether any of the patients included in the model are taking any drugs for osteoporosis?

Response: We excluded the subjects who were prescribed bisphosphonate and any of the agents which affecting bone formation. None of the subjects have prescribed bisphosphonate before stroke onset.

  1. Can you clarify the units or the levels for each of the variables used in your models? For example, are previous stroke and hypertension all modeled as two-level, yes/no variables? Also, how were all of these variables assessed? Questioning the patients or chart review?

Response: When we prospectively enrolled patients’ data in our registry database, we collected all of the patients’ several medical histories. The patient's medical history is obtained through the patient himself or the caregiver or is obtained by examining all medications the patient is taking. 

  1. Can the authors clarify in the methods and results clarify the stratification strategy? You state that the multivariate modeling is stratified by osteoporosis severity, I assume that this means that osteoporosis was separated into a two level variable of tscore<-2.5 and tscore > 2.5? 

Response: We appreciated your comment. We categorized BMD groups as low BMD group (osteoporosis group) and normal BMD group (no osteoporosis group) according to the T score. We clarified in Methods section as follows:

[Methods] [page 7, paragraph 4]

In this study, we separately analyzed the effect of BMD of the femoral neck (Model 1) and BMD of the lumbar spine (Model 2) on the outcomes. With respect to the primary and secondary outcome measures, the osteoporosis and non-osteoporosis groups were compared using Pearson’s chi-squared test, and independent effects of GA on those outcome measures were analyzed using multivariate regression analysis.

  1. Can you also include units for variables like BMI?

Response: We included BMI in all subjects in this study (Table 1). BMI was not different significantly between the two groups.

  1. The methods don't include mention of all of the variables in Table 1. For example can the author include a description of NIHSS score and stroke subtype in the methods?

Response: We appreciated your comment. NIHSS is a score representing stroke severity. We added the description of NIHSS in the Method section. In addition, the stroke subtype in Table 1 is the same as the stroke mechanism in the Method section. To avoid confusion, we revised the “Stroke subtype” and “Stroke mechanism” in both sections to the same.

[Methods] [page 5, paragraph 2]

The following data were obtained directly from the registry database: (1) demographic characteristics: age and sex; (2) stroke risk factors and medical history: previous stroke, hypertension, diabetes mellitus, hyperlipidemia, atrial fibrillation, current smoking status, and prior antithrombotic drug use; (3) clinical characteristics and acute stroke treatment: initial National Institute of Health Stroke Severity Scale (NIHSS) score which is tool to quantify the degree of impairment after stroke (the maximum possible score is 42, with the minimum score being a 0), ischemic stroke mechanism according to the TOAST criteria with some modifications (small vessel occlusion, large artery atherosclerosis, cardioembolism and others)

[Table 1]

Non osteoporosis (n=190)

Osteoporosis (n=70)

P-value

Age, years (SD)

71.5 (10.9)

76.7 (8.4)

0.06b

Female (%)

90 (47.4)

63 (90.0)

<0.001a

BMI, kg/m2 (SD)

24.4 (4.0)

22.9 (3.6)

0.90b

Initial NIHSS score (%)

0.70a

0-4

132 (69.5)

48 (68.6)

5-11

45 (23.7)

19 (27.1)

>11

13 (6.8)

3 (4.3)

Stroke mechanisms (%)

0.44a

SVO

50 (26.3)

23 (32.9)

LAA

72 (37.9)

21 (30.0)

CE

34 (17.9)

10 (14.3)

Other

34 (17.9)

16      14.3)

  1. Just to clarify, the mediation analysis test the interaction of BMD and WMD (i.e BMDxWMD)?

Response: We analyzed the mediation effect of WMD between BMD and cognitive function by using BMDxWMD in a multivariate model.

  1. What is the unit of change of BMD used in the linear regression models? It is the t-score or g/cm2?

Response: We appreciated your comment. We performed a linear regression model to evaluate the correlation between volumetric BMD (g/cm2) and the MMSE score.

We revised the Method section as follows:

[Methods] [page 8, paragraph 4]

As a sensitivity analysis, we evaluated the impact of osteoporosis on cognitive function as defined by the Montreal Cognitive Assessment (MoCA) using the van Steenovan MoCA-MMSE conversion scale. We divided the MoCA scores into three groups (MoCA score of 0~10, 11~17 and 18~30 points) 15-17. The sensitivity of the association between the volumetric BMD (g/cm2) and the K-MMSE score was investigated using a linear regression analysis. All data analyses were performed with IBM SPSS version 21.0 software (IBM Corporation, Armonk, NY, USA). Crude and adjusted odds ratios (ORs) and 95% confidence intervals (CIs) were estimated.

  1. When was WMD measured? During the acute phase, the recovery phase, or both? 

Response: WMD were measured from initial FLAIR images on brain MR. Since we performed brain MR during hospitalization, WMD was checked during the acute phase.

Results

  1. 2 "The osteoporosis group likely had more..." the results show that this is significant. Would recommend removing the term likely.

Response: We appreciated your opinion. We removed the term “likely” as your recommendation.

[Result] [page 10, paragraph 2]

The osteoporosis group had more cognitive impairment (K-MMSE score <24) than non-osteoporosis group (81.4% versus 57.4%, p<0.001). During the acute phase of ischemic stroke, the proportion of severe cognitive impairment (K-MMSE score of 0 to 17) in osteoporosis group was higher that non-osteoporosis group (47.1% versus 30.5%, p=0.001).

  1. Check the image quality of figure 2

Response: We checked the quality of Figure 2. The DPI of Figure 2 was 600.

  1. In tables 2 and 3, can the authors please label which phase of the stroke they are presenting data from? For example, I believe table 2 is in the acute phase, but the table doesn't clearly present this. 

Response: We appreciated your comment. We distinguished the ORs between acute and recovery phases of stroke in Table 2. The results of Table 3 were not dependent on the phases of stroke. The association between WMD and BMD was analyzed by Database 3 including total subjects with brain MR (See Figure 1).

Mild cognitive impairment

 vs No cognitive impairment

Severe cognitive impairment

 vs No cognitive impairment

Adjusted OR (95% CI)

in the acute phase

Adjusted OR (95% CI)

in the recovery phase

Adjusted OR (95% CI) in the acute phase

Adjusted OR (95% CI)

in the recovery phase

Total BMD

T score<-2.5a

2.64 (1.05-6.61)

2.90 (0.80-10.52)

2.22 (0.84-5.86)

2.55 (0.69-9.51)

Femoral neck BMD

T score<-2.5b

3.09 (0.87-10.99)

4.28 (0.43-42.21)

4.09 (1.11-15.14)

11.17 (1.12-110.98)

Lumbar spine BMD T score<-2.5a

1.07 (0.40-2.84)

2.94 (0.68-12.61)

0.92 (0.34-2.50)

1.53 (0.35-6.70)

  1. In table 2 it would be nice to show when the relationship is significant. It is mentioned in the text, but it if the authors could include the p-value, or give some other indication, such as bolding or adding a star or asterix, it would help the readers follow along.

Response: We appreciated your comment. We added Asterix marks on ORs which was statistically significant.

Table 2. Multivariate analysis: association between BMD and cognitive impairment (analyzed using databases 1 and 2).

Mild cognitive impairment

 vs No cognitive impairment

Severe cognitive impairment

 vs No cognitive impairment

Adjusted OR (95% CI)

in the acute phase

Adjusted OR (95% CI)

in the recovery phase

Adjusted OR (95% CI) in the acute phase

Adjusted OR (95% CI)

in the recovery phase

Total BMD

T score<-2.5a

2.64 (1.05-6.61) *

2.90 (0.80-10.52)

2.22 (0.84-5.86)

2.55 (0.69-9.51)

Femoral neck BMD

T score<-2.5b

3.09 (0.87-10.99)

4.28 (0.43-42.21)

4.09 (1.11-15.14) *

11.17 (1.12-110.98) *

Lumbar spine BMD T score<-2.5a

1.07 (0.40-2.84)

2.94 (0.68-12.61)

0.92 (0.34-2.50)

1.53 (0.35-6.70)

Abbreviations: BMD, bone mineral density; OR, odds ratio; CI, confidence interval

aAdjusted for age, female sex, years of education, diabetes mellitus status, current smoking status, low-density lipoprotein level, creatinine level and the Fazekas score

bAdjusted for age, female sex, years of education, prior stroke status, current smoking status, low-density lipoprotein level, creatinine level and the Fazekas score

* means statistically significant

  1. What is the relationship between cognition in the acute phase and recovery phase? For example, if someone is cognitively impaired in the acute phase, do they have a very high likelihood of being cognitively impaired in the recovery phase as well? If so, did the authors adjust for acute cognition in any of the models run in the recovery phase? Would it be possible to include change in cognition between acute and recovery to test whether osteoporosis or BMD affected the trajectory? 

Response: We appreciated your comment. Since it is well known that stroke itself has a great influence on cognitive function, the cognitive evaluation of the recovery phase is more reasonable than the cognitive evaluation of the acute phase of stroke. Therefore, it is considered that the acute and recovery phase of cognitive status are independent of each other. So, we investigated the effect of low BMD on cognitive function by classifying the acute phase and recovery phase to minimize the effect of stroke itself on cognitive function and described the trend of cognitive change from acute phase to recovery phase. As we comment in the Discussion section, the cognitive function was tended to be improved (proportion of severe cognitive impairment: 35.0% in acute phase and 33.5% in recovery phase). Our study aimed to determine the trend in cognitive decline during the acute and recovery phases in both the osteoporosis and non-osteoporosis groups and to compare the impact of osteoporosis on cognitive function in those two phases of stroke.

Discussion

  1. The authors refer to WMD as a potential mediator in the discussion, but it doesn't appear that Fazekas scale is associated with cognition in models including BMD (Tables s5 and s6). If it were mediating the association between BMD and cognition, wouldn't the authors first run models showing that BMD is significantly associated with cognition, then run models showing that when Fazekas score is included, the association between BMD and cognition is no longer significant or considerably attenuated? Unless I am missing something, the results presented in Tables 2 and 3 seem to suggest that BMD contributes to both WMD and cognition and the contribution of BMD to cognition is independent of WMD.  

Response: We deeply appreciate your comment. Statistically, the reviewer’s opinion was reasonable. However, at the initiation of the study, we hypothesized the WMD could modify the effect between BMD and cognition and want to find the potential causal link of BMD and cognition via WMD clinically. Also, it could be assumed that the mediation effect of WMD could be different between the early and delayed phases of stroke. So, we performed the interaction analysis by classifying the post-stroke period. Hopefully, we want you to consider this situation. With respect to the results of the mediation analysis, we cautiously postulated that WMD could potentially be attributed to osteoporosis and cognitive impairment as time passed after the stroke.

  1. Line 46 - do the authors have any specific unmeasured confounding factors that they believe would would help with the generalizability of their findings? 

Response: The specific unmeasured confounding factors which we considered to cognitive decline were physical activity before and after stroke. Physical activity could affect BMD but also cognitive function. (Jia et al. BMC geriatrics 2019;19:81, Peel et al. Journal of Gerontology 2019; 74: 943-948). Although the functional outcomes at 3month which represent the inability of independent gait (mRS >2) were collected in our database, the detailed information of physical activity, gait speed, the performance of rehabilitation was not available in our study.

  1. In the limitations, the authors note that there population was older, which is true, but they don't comment on the fact that is includes more female patients. 

Response: We appreciated your comment. Initially, since it is well known that osteoporosis was more frequent in women than men, we did not state gender disparities between two groups. In addition, we considered that sex differences could less affect cognitive function than age. Hence, we just stated the older age in this study as a limitation.

  1. The authors reference an association between osteoporosis and cognitive decline, but unless I missed it there was not time-dependent variable to get at cognitive decline, wasn't the analysis primarily cross-sectional at two time points separately?

Response: We appreciated your comment. Our study checked cognitive function at hospitalization and 1 year later after stroke. The aim of this study, as stated in the Discussion section, is to determine the trend in cognitive decline during the acute and recovery phases in both the osteoporosis and non-osteoporosis groups and to compare the impact of osteoporosis on cognitive function in those two phases of stroke. In this situation, we evaluated whether WMD has a causal link between osteoporosis and cognitive decline. We consecutively collected patients’ data and performed cognitive function test serially.

  1. I would recommend change to an association between osteoporosis and cognition.

Response: We appreciate your recommendation. We revised the Title as follows:

[Title]

TITLE: Association between Osteoporosis and Cognitive Impairment during the Acute and Recovery Phases of Ischemic Stroke

  1. The conclusions seem too broad. For example A low BMD is associated with poor cognitive function should probably include "in ischemic stroke patients.

Response: We appreciated your comment. We revised the Conclusion section.

[Conclusion] [page 17, paragraph 2]

A low BMD is associated with poor cognitive function in ischemic stroke patients.

Reviewer 2 Report

  • A brief summary (one short paragraph) outlining the aim of the paper and its main contributions.

There is evidence of a link between low bone mass and cerebrovascular disease, but this area of research is not well explored. The authors performed a retrospective clinical study examining the relationship between cognitive function and bone density in the lumbar spine and right femoral neck following ischemic stroke. They found that low bone mineral density was associated with worse cognitive function immediately following stroke and a year following stroke. The study is the first to make this connection, and provides important considerations for clinicians, rehabilitation workers, and future research into the connection between BMD and stroke.

  • Broad comments highlighting areas of strength and weakness. These comments should be specific enough for authors to be able to respond.

- Overall this paper is very well founded, supported, and written. I found the results interesting and believe they have important clinical implications even if no causative or mechanistic explanations are present.

- Very interesting results in the supplementary tables. The large discrepancy between osteoporosis groups’ smoking and diabetes incidence is extremely interesting. Stroke severity and functional outcome similar b/w groups is good to see. I like that you addressed physical activity in the discussion.

- My only major concern with the paper is the lack of discussion of BMD loss due to bed rest following stroke, which could be a major confounding factor to the analysis of the results and the conclusions the authors make. I do not think that redoing the study can overcome this ‘chicken or the egg’ problem, so I think addressing this issue in the discussion would be enough. See my specific comments below. Similarly, only measuring BMD in the right femur may be confounded based on which side of the patient’s body is affected by the stroke, which has been shown to lose BMD more quickly than the non-paretic side.

- A second concern is the lack of transparency in the timepoints of the BMD, neurological damage, and cognitive function measurements. During the acute recovery period bone mass is rapidly lost while cognitive function rapidly recovers. The methods and/or results should include explanations of exactly when BMD was assessed following stroke (if known).

  • Specific comments referring to line numbers, tables or figures. Reviewers need not comment on formatting issues that do not obscure the meaning of the paper, as these will be addressed by editors.

Introduction

  • Could be a little longer. Some of the points made in the first 2 paragraphs of discussion could be moved to introduction. This is just a suggestion, I found the introduction easy to follow and understood your hypothesis and rationale. Maybe discuss the importance of the acute recovery period here.

Methods

  • It would be helpful to include when the different assessments took place during stroke recovery. It is known that stroke patients lose bone mass rapidly during the acute recovery phase while simultaneously rapidly regaining cognitive function. Including the timing of these outcome measures, and possibly examining how they change during recovery, would elevate the impact of your results.
  • Figure 1 caption should be expanded to briefly explain exclusion criteria for database 1 and 2.
  • Brief description of what mRS scores mean would be helpful. (i.e. 2 = slight disability, 3 = moderate disability, etc).
  • The groups of the study are confusing and should be outlined in more detail (i.e. there are 3 database groups, but then an ‘osteoporosis’ group, a lumbar spine group, and a femoral neck group). What is the overlap between these groups? Was osteoporosis in the ‘osteoporosis’ group self-reported or was it also determined from the spine and femur BMD? Just needs to be cleared up. It seems clear in Table 2, but this should be explained in methods.

Results

  • Again, defining the recovery timepoints would strengthen the results
  • First sentence of section 3.3 should have p-value
  • The last paragraph is extremely interesting result!
  • If possible, authors should try to compare the femoral BMD to cognitive function based on whether the left or right side of the patient is paretic. There is evidence that bone is preferentially lost on the paretic side, which could confound the Interpretation of this study, and potentially hide an interesting result.

Discussion

  • Sentence line 43-45 has a grammar / typo issue.
  • Might be worth mentioning again that you’ve controlled for sex.
  • Increased cognitive impairment leads to more bedrest and less use of the limbs, which will lead to rapid loss of bone mass. Since the lumbar spine is less mechanically sensitive, this may explain some of your results. This is a ‘chicken or the egg’ problem that cannot be fully addressed without knowing pre-stroke BMD. This needs to be addressed in the discussion. Again, examining if the patients’ right or left side is paretic could help mitigate this issue with your interpretation of the results.

You mention in line 75 that the study was prospective but in abstract it says retrospective.

Author Response

Reviewer#2

Specific comments referring to line numbers, tables or figures. Reviewers need not comment on formatting issues that do not obscure the meaning of the paper, as these will be addressed by editors.

Introduction

  1. Could be a little longer. Some of the points made in the first 2 paragraphs of discussion could be moved to introduction. This is just a suggestion, I found the introduction easy to follow and understood your hypothesis and rationale. Maybe discuss the importance of the acute recovery period here.

Response: We appreciated your comment. As reviewer’s recommendation, we have revised the Introduction section as follows:

[Introduction] [page 4 paragraph 1]

Since ischemic stroke can result in physical disability and cognitive impairment 5 and 40% of patients with stroke present with preexisting osteoporosis 6, a study designed to evaluate the effect of osteoporosis on cognitive impairment after ischemic stroke would be interesting. Cognitive function is known to be influenced by the time of stroke occurrence. Hence, the treatment strategy of stroke may vary depending on the early and delayed periods of stroke.

Recently, BMD was reported to be associated with microvascular diseases, such as cerebral white matter disease (WMD); a lower BMD indicates a more severe WMD 7-8. Additionally, WMD is a well-known risk factor for dementia in several studies 9. Hence, we postulated that low BMD may be associated with cognitive impairment.

Methods

2.It would be helpful to include when the different assessments took place during stroke recovery. It is known that stroke patients lose bone mass rapidly during the acute recovery phase while simultaneously rapidly regaining cognitive function. Including the timing of these outcome measures, and possibly examining how they change during recovery, would elevate the impact of your results.

Response: We appreciated your comment. Bone loss is accelerated during the first year after stroke (Jorgensen L. et al. Bone.2001;28:655-9) and the risk for post-stroke cognitive impairment is known to be highest within the first year(Pendlebury ST et al. Lancet Neurology.2009;8:1006-18). Hence, we assessed the BMD at the early phase of stroke and cognitive function test at 1-year after stroke. We have revised the Methods section as follows:

[Methods][page 5, paragraph 1]

Because patients with severe neurological deficits were unable to perform or refuse cognitive function tests, we established three separate databases: database 1 excluded subjects who were not able to complete the Korean Mini-Mental State Examination (K-MMSE) evaluating the effect of osteoporosis on cognitive function during hospitalization; database 2 excluded subjects who were not able to complete the K-MMSE at 1 year after stroke onset; and database 3 included all eligible subjects with WMD (Figure 1). Bone loss is accelerated during the first year after stroke and the risk for post-stroke cognitive impairment is known to be highest within the first year. Hence, we assessed the BMD at the early phase of stroke and cognitive function test at hospitalization and 1-year after stroke

3.Figure 1 caption should be expanded to briefly explain exclusion criteria for database 1 and 2.

Response: We appreciated your opinion. We have revised the Figure legends as follows;

FIGURE LEGENDS

Figure 1. Flow chart of the study.

Database 1: excluding subjects without MMSE at hospitalization

Database 2: excluding subjects without MMSE at 1 year after stroke

Abbreviations: MMSE, Mini-Mental State Examination

4.Brief description of what mRS scores mean would be helpful. (i.e. 2 = slight disability, 3 = moderate disability, etc).

Response: We appreciated your comment. We have revised the Methods section as follows

[Methods] [page 6, paragraph 1]

(5) functional status defined by the mRS score (0=no symptoms; 1=no significant disability; 2=slight disability, able to look after own affairs without assistance; 3=moderate disability, requires some help, but able to walk unassisted; 4=moderately severe disability, unable to attend to own bodily needs without assistance; 5=severe disability, requires constant nursing care and attention, bedridden; 6=dead) at discharge and at 3 months.

5.The groups of the study are confusing and should be outlined in more detail (i.e. there are 3 database groups, but then an ‘osteoporosis’ group, a lumbar spine group, and a femoral neck group). What is the overlap between these groups? Was osteoporosis in the ‘osteoporosis’ group self-reported or was it also determined from the spine and femur BMD? Just needs to be cleared up. It seems clear in Table 2, but this should be explained in methods.

Response: We appreciated your comments. Osteoporosis group was a combination of the femur neck osteoporosis group or/and the lumbar spine osteoporosis group. The “osteoporosis group” was determined from the BMD T score <-2.5 of the femur neck and lumbar spine. We have revised the Method section as follows;

[Methods] [page 8, paragraph 3]

In this study, we separately analyzed the effect of BMD of the femoral neck (Model 1) and BMD of the lumbar spine (Model 2) on the outcomes. The osteoporosis group was a combination of the BMD T score <-2.5 of the femur neck and/or lumbar spine.

Results

6.Again, defining the recovery timepoints would strengthen the results

Response: We have added the definition of the recovery phase as follows;

[Results] [page 11, paragraph 2]

The multivariate analysis revealed a correlation between osteoporosis and mild cognitive impairment in the acute phase, but not with severity of cognitive impairment during the recovery phase (defined as 1-year after stroke) of ischemic stroke.

7.First sentence of section 3.3 should have p-value

Response: We have added the p-values as follows:

Osteoporosis of the femoral neck was associated with a severe cognitive impairment (adjusted OR 4.09, [95% CI] 1.11-15.14, p=0.035 in the acute phase; adjusted OR 11.17, [95% CI] 1.12-110.98, p=0.04 in the recovery phase), whereas osteoporosis of the lumbar spine was not associated with cognitive impairment.  

8.The last paragraph is extremely interesting result!

Response: We deeply appreciated your interest.

9.If possible, authors should try to compare the femoral BMD to cognitive function based on whether the left or right side of the patient is paretic. There is evidence that bone is preferentially lost on the paretic side, which could confound the interpretation of this study, and potentially hide an interesting result.

Response: We deeply agreed with your comments. However, the femur neck BMD was measured on only the right side in our institution. After our study, we started to measured BMD on both sides of the femur neck. Our next work could evaluate the impact of femur neck BMD of paretic/non-paretic sites on several outcomes.

Discussion

10.Sentence line 43-45 has a grammar / typo issue.

Response: We have revised the sentence as follow;

However, since both femoral neck and lumbar spine BMD were associated with poor outcome in previous analysis of cerebrovascular disorders, and other studies measured only the femur neck BMD to investigate stroke risk, we should use caution when generalizing these results.

11.Might be worth mentioning again that you’ve controlled for sex.

Response: We have mentioned again that adjusting for sex in statistical analysis as a reviewer’s recommendation;

Since we adjusted for age and sex in all multivariate models and found no difference in GDS scores between the osteoporosis group and the non-osteoporosis group, we assume that the impacts of age and depression on the outcomes have been attenuated in this study.

  1. Increased cognitive impairment leads to more bedrest and less use of the limbs, which will lead to rapid loss of bone mass. Since the lumbar spine is less mechanically sensitive, this may explain some of your results. This is a ‘chicken or the egg’ problem that cannot be fully addressed without knowing pre-stroke BMD. This needs to be addressed in the discussion. Again, examining if the patients’ right or left side is paretic could help mitigate this issue with your interpretation of the results.

You mention in line 75 that the study was prospective but in abstract it says retrospective.

Response: We deeply appreciated your comment. We have revised the Discussion section as follows;

Also, this study is a “retrospective” observational study, but patients’ information was consecutively collected in the registry database in our institution.

[Discussion] [page 15, paragraph 1]

However, since both femoral neck and lumbar spine BMD were associated with a poor outcome in the previous analysis of cerebrovascular disorders 2, and other studies measured only the femur neck BMD to investigate stroke risk 1, 3, we should use caution when generalizing these results. Also, pre-stroke BMD was not available in our study and the lumbar spine has known to be less sensitive to mechanical stimulation. Hence, we should be careful about clinically interpreting this result. Nonetheless, these results could prompt researchers to identify a reasonable BMD site for predicting cognition in patients with stroke. Further studies are needed to clarify this issue.

[Discussion] [page 15, paragraph 3]

Although this study is the first to show an association between a low BMD and cognition in patients with stroke, it has several limitations. First, this study was a single-center study with a relatively small number of subjects, despite the use of a registry database consecutively collecting patients’ data.
